# Identification of Factors Affecting the Increased Percentage of CGA Recommendations among Patients on Geriatric Ward

**DOI:** 10.3390/ijerph20032065

**Published:** 2023-01-23

**Authors:** Justyna Gołębiowska, Małgorzata Szymala-Pędzik, Joanna Żórawska, Małgorzata Sobieszczańska, Siddarth Agrawal

**Affiliations:** 1Department and Clinic of Internal Medicine, Occupational Diseases, Hypertension and Clinical Oncology, Wroclaw Medical University, 50-556 Wroclaw, Poland; 2Clinical Department of Geriatrics, Wroclaw Medical University, Pasteur 4 Street, 50-367 Wroclaw, Poland

**Keywords:** geriatric assessment, CGA recommendation, frailty syndrome, factors of CGA recommendation

## Abstract

In Poland, the elderly population is noticeably increasing every year. Therefore, the current healthcare system has to rise to the challenge of treatment and prevention strategies targeting elderly persons. Based on the Vulnerable Elders Survey (VES-13 scale), consisting of international and validated scales, we put effort into identifying the factors affecting the increased percentage of CGA (Comprehensive Geriatric Assessment) recommendations by healthcare system physicians. The study group involved 78 patients from the Department of Geriatrics, Wroclaw Medical University, Poland, aged 60–91 (median Me = 81 years old). Of the studied patients, 51 were recommended for CGA (51/78; 64.6%). A statistically significant association was observed between CGA recommendation and age (*p* < 0.001); the likelihood of a CGA recommendation increased with age. Furthermore, the increased CGA recommendation was observed among geriatric patients with: (a) frailty syndrome (OR = 11.2, CI95% 2.88–43.5, *p* < 0.001), (b) high risk of malnutrition or malnutrition (OR = 3.87; CI95%, 1.00–14.9, *p* = 0.04), (c) low mental status (OR = 3.32, CI95% 1.22–9.09, *p* = 0.029), (d) low ability to perform basic daily duties (according to ADL scale—OR = 12.6, CI95% 1.57–101, *p* = 0.004 and IADL scale—OR = 4.71, CI95% 1.72–12.9, *p* = 0.003), and (e) fall risk (OR = 15.0, CI95% 4.67–48.3, *p* < 0.001). Moreover, homocysteine levels increased with age (*p* = 0.003) and decreased with sleep duration (*p* = 0.043). Taken together, all these factors seem to be important when implementing treatment programs adjusted to the individual geriatric patient.

## 1. Introduction

The Polish population is aging. In 2016, over 6 million patients aged 65+ in Poland constituted approximately 16% of the total population [1]. It is estimated that this percentage will double in the next 20 years. According to the European Commission, the Polish population, along with the Latvian and Romanian ones, will make these among the oldest European Union countries [2]. As defined by the European Union of Medicine in Assurance and Social Security (EUMASS), a typical geriatric patient is over 70 years, with multimorbidity and the highest risk of different complications, potentially leading to the “domino” effect, whereby one adverse event provokes the occurrence of another. Its consequences seem to include the loss of autonomy, physical efficiency, cognitive function, and even death [3].

An accurate estimation of the overall survival (OS) of geriatric patients may be performed by clinical information available from the medical chart or a brief interview with the patient or his/her caregiver. There are also available predictive measures to identify, quantify, and manage the health condition of geriatric patients. The comprehensive geriatric assessment (CGA) is a gold standard to determine a frail older person’s psychological and functional ability to develop a coordinated and integrated plan for treatment and long-term follow-up [4]. This tool includes different validated and optimized geriatric scales or tests to prepare an overview of the patient’s health, which may be useful for developing an individualized geriatric intervention plan. Its beneficial advantages were proved in studies focused on elderly patients with cancer; as a tool to optimize cancer treatment selection, it improved the chances of treatment completion, overall survival, and quality of life and decreased the risk of hospitalization and nursing home placement [5,6,7].

The CGA is applied mainly to patients in old age, those with frailty syndrome, and those in critical moments of their life (e.g., after a stroke, heart attack, sudden deterioration of their economic situation, widowhood, or other stressful situations). The crucial elements of the CGA include (1) physical health assessed by medical evaluation and the assessment of disease severity, (2) functional efficiency assessed by Activities of Daily Living (ADLs scales), Tinetti Gait and Balance Assessment Tool and activity levels, (3) mental health assessed by cognitive assessment tests and mood disorder rating scales, and (4) social and environmental conditions (Figure 1) [2].

Furthermore, there are also geriatric screening tools available that combine all components, as shown in Figure 2, and that are required to select patients who would benefit from CGA, such as Geriatric 8 (G8), Vulnerable Elders Survey 13, and the Flemish version of the Triage Risk Screening Tool (fTRST). Some of these screening tools can also provide important information about treatment-related toxicity, the risk of functional decline, and overall survival [8,9,10]. These tools are based on different scales intended to determine the status of each CGA component for the patient. For instance, to demonstrate primary and psychosocial functions and activities of a patient’s daily living, the Katz Activities of Daily Living Scale [11], the Lawton Instrumental Activities of Daily Living Scale [12], the Barthel index [13], and the Tinetti test [14] may be applied. Moreover, regarding mental assessment, all available scales may be divided into three categories based on the application: (1) cognitive usage, including the mini-mental state examination (MMSE) [15] and the Montreal Cognitive Assessment (MoCA) [16], as well as the Abbreviated Mental Test score (AMTS) [17] and the Clock-Drawing Test (CDT) [18]; (2) emotional state, including the Geriatric Depression Scale (GDS) [19]; and (3) qualitative disorders of consciousness, such as the Confusion Assessment Method (CAM) [20] or the Delirium Observation Scale (DOS) [21]. Based on these questionnaires, clinical and healthcare professionals can obtain a general overview of a patient’s mental status, and they may rate their depression level and assess their potential disturbing cognitive impairments and signs of dementia [12].

In the present study, we aimed to determine the factors affecting the increased percentage of CGA recommendations by healthcare system physicians. This analysis was mediated using the Vulnerable Elders Survey (VES-13 scale). Understanding the factors increasing the percentage of CGA recommendations among geriatric patients seems to be crucial for health professionals, whose role is to control the aging processes and implement treatment strategies for comorbid diseases, as well as to encourage geriatric patients to change their lifestyle and nutrition habits.

## 2. Materials and Methods

### 2.1. Study Group

The study group involved 78 patients from the Department of Geriatrics, Wroclaw Medical University, Poland, including 65 women (65/78; 83.3%) aged 60–91 years old (median Me = 81 years old, lower quartile Q1 = 77 years old, upper quartile Q3 = 85 years old) (Table 1). The inclusion criteria were as follows: (1) aged 18 years or older; (2) a geriatric diagnosis; (3) able to read and write in Polish. The study protocol was approved (approval code: 548/2020, approval date: 7 July 2020) by the Bioethics Committee at Wroclaw Medical University. All patients provided written informed consent related to participation in the study, which was conducted according to the Declaration of Helsinki.

### 2.2. Study Design

The characteristics of all studied patients are presented in Table 1. A cross-sectional study was carried out from November–December 2020 in Poland. The evaluated sample of the geriatric population was obtained by a stratified sampling per voivodeship demographic structure of Poland. The same formula, as shown in our previous study [22], was used to calculate the proper size of the sample. Target quotas were set for age and gender in each of the geographical regions. All the participants were interviewed by computer-assisted telephone calls. The identity of a participant was confirmed at the beginning of the interview. Interviewers were adequately trained and prepared to ensure the equal and adequate quality of the survey. Moreover, all interviews were supervised by a specialist. Additionally, a study coordinator evaluated the recorded conversations. The transcripts were not returned to participants for any comment and/or correction, nor were repeat interviews carried out. The duration of the interview ranged from 15 to 20 min. Participants provided their verbal consent at the beginning of the interview and were informed about the goal of the survey. No compensation was provided for participating in the study.

### 2.3. Explanatory Variables

The VES-13 (Vulnerable Elders Survey) scale was used to identify patients requiring a CGA recommendation. If the result of the VES-13 scale shows more than three points, it is very likely to signify a significant deterioration in health, functional capacity, or death within the next two years. The CGA procedure consisted of (1) the risk of pressure ulcers by the Norton scale, (2) the risk of malnutrition by MNA or NRS scales, (3) the Geriatric Depression Scale (GDS), (4) the mini-mental state examination (MMSE), (5) the Clock-Drawing Test, (6) frailty syndrome assessment (TFI), (7) fall risk by the Tinetti test, (8) the Activities of Daily Living (ADL scale), (9) the Lawton Instrumental Activities of Daily Living Scale (IADL), and (10) the Barthel scale.

## 3. Results

Of the 78 patients studied, 51 were recommended for CGA (51/78; 64.6%, Table 1).

There was no statistically significant relationship between the survival of geriatric patients and gender, age, and recommendation for CGA (*p* > 0.05). The correlation between the assessment of the necessity to carry out a CGA and gender, age, and patient status are summarized in Table 2. A statistically significant correlation was observed between recommendation for CGA and age (*p* < 0.001); the likelihood of a CGA recommendation increases with age. Based on the ROC curve analysis, the age of 79 years is a cut-off value (Appendix A).

Among patients aged 79+, the chance of obtaining a recommendation for CGA is fifteen times higher compared to patients under 79 years old (95% CI (4.67–48.3); OR = 15.0, *p* < 0.001, Table 3).

In the analyzed group, the recommendation for CGA was obtained by 51 patients (64.6%); therefore, not all patients were assessed with the use of subsequent scales. Table 4 presents the results of patients’ assessments using the remaining tests (questionnaires). A statistically significant correlation at the level of *p* < 0.05 was observed only between the recommendation for CGA and the level of the frailty syndrome (10/49; *n* = 33.3%, *p* = 0.036, Table 4). Patients with this syndrome, regardless of the severity of this condition, were more often recommended for CGA.

For parameters significant at *p* < 0.2, a dichotomous division was performed (based on the analysis of ROC curves, Appendix A); the results are presented in Table 5. The likelihood for CGA recommendation is statistically significant in groups of patients with a high risk of malnutrition or malnutrition (OR = 3.87; CI95%, 1.00–14.9, *p* = 0.04), a mini-mental status test (MMST) < 27 points (OR = 3.32, CI95% 1.22–9.09, *p* = 0.029), and an assessment of frailty syndrome ≥ 4 points (OR = 11.2, CI95% 2.88–43.5, *p* < 0.001). Furthermore, patients for whom the ADL and IADL scales were detected as less than 6 points and less than 27 points, respectively, were more likely to be recommended for CGA (OR = 12.6, CI95% 1.57–101, *p* = 0.004 for ADL scale and OR = 4.71, CI95% 1.72–12.9, *p* = 0.003 for IADL scale). A significant statistical correlation was also determined among patients with a fall risk assessed as less than 8 points (OR = 15.0, CI95% 4.67–48.3, *p* < 0.001) and a Barthel scale with less than 100 points (OR = 3.28, CI95% 1.21–8.91, *p* = 0.031, Table 5).

Moreover, a statistically significant correlation was observed between the level of homocysteine and sleep duration (negative, *p* = 0.043, Figure 2A) and age (positive, *p* = 0.003, Figure 2B). Thus, the level of homocysteine increases with age and decreases with sleep duration. Furthermore, it is enhanced among patients with moderate and strong frailty syndrome (*p* = 0.004, Figure 2C).

## 4. Discussion

Over the last hundred years, life expectancy has increased linearly by three months per year [23], leading to an increase in the elderly population worldwide. Due to diminished homeostatic reserves, older adults are more vulnerable to iatrogenic complications that are not typical of the underlying presentations [24]. A comprehensive geriatric assessment (CGA) is one of the cornerstones of modern geriatric care that is used to develop a coordinated and integrated plan for treatment and long-term follow-up. It consists of various scales and tests to assess independence in daily functioning and the potential occurrence of specific geriatric difficulties, including falls, eating disorders, and frailty syndrome. Furthermore, its goal is to delineate the strengths of each individual, which may become a starting point for improving their functioning [25]. Our study revealed that from a total of 78 patients analyzed, 51 of them were recommended for CGA. A statistically significant correlation was observed between recommendation for CGA and age (*p* < 0.001); the likelihood of a CGA recommendation increases with age. Among patients aged 79+, the chance of obtaining a recommendation for CGA is fifteen times higher compared to patients under 79 years old (95% CI (4.67–48.3); OR = 15.0, *p* < 0.001, Table 3). The effects of these recommendations have been widely analyzed in the literature [26,27,28,29]. The role of age in CGA recommendation was found by Chen et al., who determined that positive effects of CGA interventions observed by the improvement of life quality were more likely to occur in patients older than 80 years old [30].

Apart from giant geriatric syndromes (GS) (such as dementia, delirium, sarcopenia, falls, and sphincter incontinence) occurring substantially in advancing age, frailty syndrome became a key topic of interest, both in terms of clinical practice and scientific research. It is a physiological syndrome characterized by reduced immunological reserves and resistance to stressors. It is associated with an increased risk of complications, such as loss of independence, the need for more frequent hospitalization, susceptibility to more frequent illness, or even death. Increased levels of inflammatory markers, age-related coagulation disorders, changes in the endocrine system, and loss of muscle mass or sarcopenia appear to be closely related to the development of this syndrome. The clinical criteria for diagnosis are weight loss, asthenia, exhaustion, slower walking, and decreased physical activity [31,32,33,34,35]. Our study revealed a statistically significant association at the level of *p* < 0.05 between the recommendation for CGA and the level of the frailty syndrome (10/49; *n* = 33.3%, *p* = 0.036, Table 4). Therefore, regardless of whether in the initial or intensified stages, geriatric patients with this syndrome are more likely to be recommended for CGA. This result is supported by other studies describing the evidence of associations between frailty and CGA [10,31].

It is well known that frailty syndrome is a manageable condition that can be targeted for intervention [33,34,35]. The most straightforward strategies for its treatment are exercises and caloric and protein (1.2–1.5 g/kg/day) support. Protein-calorie supplementation is effective in increasing muscle mass and grip strength. Our study revealed that the likelihood of CGA recommendation is statistically significant in a group of patients with a high risk of malnutrition or malnutrition (OR = 3.87; CI95%, 1.00–14.9, *p* = 0.04). This condition leads to an impairment of physical and mental health and adversely affects the treatment outcome of underlying disease treatment. Consistently, Cailett et al. identified the factors associated with changes in cancer treatment, including lower ADL scores ((OR), 1.25 per 0.5-point decrease; CI, 1.04 to 1.49; *p* = 0.016) and malnutrition (OR, 2.99; CI, 1.36 to 6.58; *p* = 0.007). Our study also revealed this association between low ADL scores, and hence functional impairment and more frequent CGA recommendations (OR = 12.6, CI95% 1.57–101, *p* = 0.004) [6]. The role of CGA in cancer treatment was also assessed by Zhang et al., who found that malnutrition was significantly associated with frailty. Patients with malnutrition were approximately four times more likely to have frailty syndrome (OR = 3.82, 95% CI: 1.35–10.84, *p* = 0.01) than those without malnutrition [36]. As to our study, this is another indirect association worth considering, which connects different components and creates a network of associations. Furthermore, compared to the other studies in this field, our study shows a broader concept of the recommendation of CGA, as it does not focus on specific diseases but describes the problems of geriatric patients that occur as a natural aging process.

With age, the correct function of many organs, including the kidneys, declines. This leads to a decrease in homocysteine secretion and hence its rise in the plasma. Such a result was found in our study. The level of homocysteine increases with age (positive correlation, *p* = 0.003, Figure 2B). Furthermore, it is enhanced among patients with moderate and strong frailty syndrome (*p* = 0.004, Figure 2C). Homocysteine causes the thickening and narrowing of both the coronary and cerebral arteries. This protein also damages blood vessels by potentially reducing the supply of oxygen and nutrients to the brain. It is considered a reliable biochemical marker for cardiovascular diseases [22,37,38], neurocognitive disorders [39,40], and osteoporotic fractures [41,42]. In line with our study, Xu et al., in a group of 7872 individuals, demonstrated that a high level of homocysteine was associated with more frequent hospitalization, especially in geriatric patients. It increases with age in both genders (especially in postmenopausal women, which is associated with a decrease in estrogen levels) [43].

CGA can be effective in managing geriatric syndromes only if its recommendations are followed; the most crucial role belongs to primary health professionals, whose role is to implement these recommendations in the most affordable way. To obtain the most reliable information about each geriatric patient, the CGA scales should be precisely chosen and carefully analyzed to formulate a specific treatment and/or prevention plan. A patient’s caregivers are strongly supportive in assessing the functional status of geriatric patients. However, this should not be a substitute for a conversation with a patient, which allows for the assessment of functional status based on general observation and potentially catching the source of mental or clinical diseases that have not been considered before. Therefore, it is highly beneficial to acquire the correct language in patient-doctor relationships to carry out an examination that fully reflects the actual health status of a patient. Thanks to a careful survey, we can assess patients’ priority goals. It is worth noting that functional and social abilities may exceed the need for improving health and extending the span of life [44]. These tests are mainly necessary for nursing homes or retirement homes, where one person’s mental state impacts other residents [29,45].

Due to the increased number of older adults and the lack of healthcare professionals who specialize in this demographic, the scarcity of geriatricians is a severe problem in public health worldwide. Currently, the availability of specialist geriatric care in Poland is insufficient. It includes 518 doctors specializing in geriatrics [46]. In comparison, in the United States (US), there are approximately 7000 geriatricians (data from July 2022), and around half of them are full-time [47]. In Russia, approximately 300 full-time geriatricians were employed by the end of 2018, and in Turkey, there were only 57 licensed geriatricians in 2021 [48]. In the Polish healthcare system, care for elderly patients is carried out within the framework of primary healthcare in cooperation with a network of geriatric specialists. Numerous programs are implemented for the early diagnosis, treatment, and prevention of diseases of old age. Doctors, nurses, physiotherapists, as well as medical caregivers and community therapists participating in these programs, gain knowledge and skills that are useful in daily practice. Nevertheless, updates are still required, especially these days, when the COVID-19 pandemic is still taking its toll and significantly impacts older people’s mental and clinical health [49,50,51,52]. Our study is such an update and may be used to implement new programs, thanks to which healthcare professionals learn the proper methods for dealing with specific clinical situations. Coordinated care, supported by a thorough scale, can presumably bring further benefits to the patient. These procedures seem to improve the quality of life of the elderly and their family caretakers. Comprehensive long-term care can be planned and organized, and the willingness and personal dignity of older people can be respected.

Several limitations to this study should be borne in mind. Independent factors, such as the time of day or a patient’s mood, may impact the answers needed to assess the VES-13 scale. Some misclassifications might have occurred because the diagnosis of geriatric syndromes was based on self-reported data rather than clinical examinations. Furthermore, patients were not tested according to the source of the disease or the level of severity of the disease, and the patient’s medical history was not taken into account. Thus, CGA was performed before treatment in some patients and during treatment in others, sometimes after several lines of therapy (surgery, chemotherapy). It is also worth considering the group sizes for which the statistical significance has been noticed. Due to the small size of some groups, it is possible to obtain a false positive result. It is also quite challenging to assess the definition of frailty; for each study investigator, frailty may be interpreted slightly differently.

## 5. Conclusions

This study revealed, for the first time in Poland, the factors increasing the likelihood of recommending the CGA: (1) the presence of frailty syndrome, (2) older age, (3) malnutrition and risk of malnutrition, (4) low mental status, (5) problems with performing daily duties (according to ADL and IADL scales), and (6) a high level of homocysteine. Therefore, we suggest taking into account the scales and screening tools that allow clinicians to determine patients’ CGA components, as mentioned above, and assess their health status. The current literature focuses on patients with cancer treatments; our study revealed a broader spectrum, including patients with different diseases and health impairments. Furthermore, our findings show that the implementation of mandatory CGA assessment among patients older than 79 should be considered in Poland.

## Figures and Tables

**Figure 1 ijerph-20-02065-f001:**
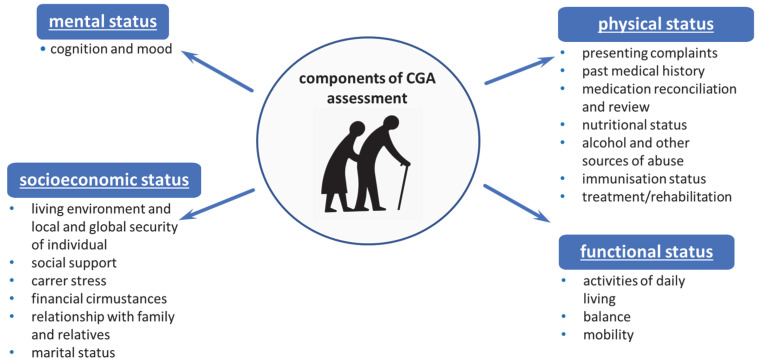
The four main components covered in a CGA assessment.

**Figure 2 ijerph-20-02065-f002:**
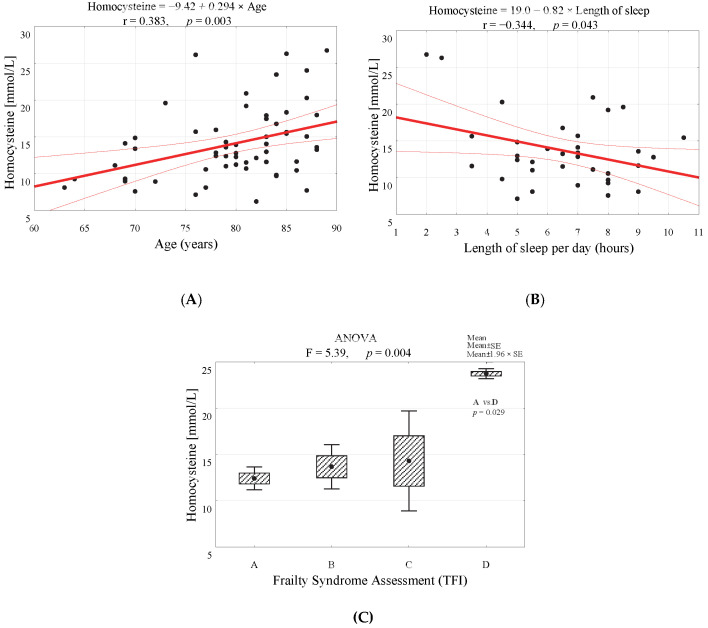
Correlation diagrams of homocysteine level with (**A**) age and (**B**) sleep duration. (**C**) The level of homocysteine in groups differing in the level of the frailty syndrome (A—infrangible, B—frangible, C—slightly frangible, D—moderately or strongly frangible) and the result of the analysis of variance.

**Table 1 ijerph-20-02065-t001:** Overall characteristics of patients.

Feature (Variable)	All Patients *N* = 78	Patient Status		*p*
		Still Living (as of 10 October 2022) *N* = 69	Dead *N* = 9	
Gender				0.660
Women, *n* (%)	64 (82.1)	57 (82.6)	7 (77.8)	
Men, *n* (%)	14 (17.9)	12 (17.4)	2 (22.2)	
Age (years old)				0.199
Age 60–74 years old, *n* (%)	14 (17.9)	14 (20.3)	0 (0.0)	
Age 75–84 years old, *n* (%)	44 (56.4)	39 (56.5)	5 (55.6)	
Age 85+ years old, *n* (%)	20 (25.6)	16 (23.2)	4 (44.4)	
Identification of patients requiring CGA				0.485
Lack of recommendation, *n* (%)	27 (34.2)	25 (36.2)	2 (22.2)	
Recommendation for CGA, *n* (%)	51 (64.6)	44 (63.8)	7 (77.8)	

*n*—sample size, %—ratio, *p*—significance level.

**Table 2 ijerph-20-02065-t002:** General characteristics of patients in groups differing in the risk assessment of significant worsening health, functional fitness, or death within the next 2 years (VES-13 scale).

Feature (Variable)	VES-13 Scale		*p*
	<3 Points *N* = 27	≥3 Points *N* = 51	
Gender			0.685
Women, *n* (%)	21 (77.8)	43 (84.3)	
Men, *n* (%)	6 (22.2)	8 (15.7)	
Status			0.485
Living, *n* (%)	25 (92.6)	44 (86.3)	
Dead, *n* (%)	2 (7.4)	7 (13.7)	
Age (years old)			<0.001
Age 60–74 years, *n* (%)	11 (40.7)	3 (5.9)	
Age 75–84 years, *n* (%)	16 (59.3)	28 (54.9)	
Age 85+ years, *n* (%)	0 (0.0)	20 (39.2)	

**Table 3 ijerph-20-02065-t003:** Number (percentage) of patients in groups differing in age and recommendation for CGA, the result of the independence test (*p*), and the odds ratio and its 95% confidence interval.

Age (Years Old)	Recommendation for CGA		*p*	OR [95% CI]
	Yes (VES-13 ≥ 3 Points) *N* = 51	No (VES-13 < 3 Points) *N* = 27		
79 and more	45 (88.2)	9 (33.3)	<0.001	15.0 [4.67; 48.3]
to 78	6 (11.8)	18 (66.7)		1.00 [ref.]

**Table 4 ijerph-20-02065-t004:** Number (percentage) of all patients in groups differing in age and recommendation for CGA, the result of the independence test (*p*), and the odds ratio and its 95% confidence interval.

	All Studied Patients	CGA Recommendation		*p*
		Yes	No	
Pressure ulcers risk (Norton scale):				0.265
No risk, *n* (%)	3 (4.7)	3 (7.9)	0 (0.0)	
High risk, *n* (%)	61 (95.3)	35 (92.1)	26 (100.0)	
Risk of malnutrition (MNA or NRS):				0.097
Low risk of malnutrition, *n* (%)	55 (75.3)	31 (67.4)	24 (88.9)	
High risk of malnutrition, *n* (%)	15 (20.6)	12 (26.1)	3 (11.1)	
No malnutrition, *n* (%)	3 (4.1)	3 (6.5)	0 (0.0)	
Geriatric Depression Scale (GDS):				0.386
No depression, *n* (%)	64 (85.3)	40 (81.6)	24 (92.3)	
Mild depression, *n* (%)	9 (12.0)	7 (14.3)	2 (7.7)	
Major depression, *n* (%)	2 (2.7)	2 (4.1)	0 (0.0)	
Mini-mental state examination (MMSE):				0.096
No cognitive impairment, *n* (%)	39 (52.0)	20 (41.6)	19 (70.4)	
Cognitive impairment without dementia, *n* (%)	13 (17.3)	9 (18.8)	4 (14.8)	
Mild dementia, *n* (%)	11 (14.7)	9 (18.8)	2 (7.4)	
A moderate degree of dementia, *n* (%)	12 (16.0)	10 (20.8)	2 (7.4)	
Clock-Drawing Test (points):				0.305
Correct clock drawing, *n* (%)	18 (24.3)	10 (20.8)	8 (30.8)	
Impaired clock drawing, *n* (%)	56 (75.7)	38 (79.2)	18 (69.2)	
Frailty syndrome assessment (TFI):				0.036
Infrangible, *n* (%)	28 (57.1)	12 (40.0)	16 (84.2)	
Frangible, *n* (%)	11 (22.5)	10 (33.3)	1 (5.3)	
Mildly frangible, *n* (%)	7 (14.3)	5 (16.7)	2 (10.5)	
Moderately frangible, *n* (%)	1 (2.0)	1 (3.3)	0 (0.0)	
Strongly frangible, *n* (%)	2 (4.1)	2 (6.7)	0 (0.0)	
Fall risk:				0.123
Low, *n* (%)	28 (44.5)	14 (35.0)	14 (60.9)	
Moderate, *n* (%)	13 (20.6)	9 (22.5)	4 (17.4)	
High, *n* (%)	22 (34.9)	17 (42.5)	5 (21.7)	
Activities of Daily Living (ADL scale)				0.120
Completely preserved, *n* (%)	69 (90.8)	42 (85.7)	27 (100.0)	
Moderate disability, *n* (%)	4 (5.3)	4 (8.2)	0 (0.0)	
Strong disability, *n* (%)	3 (3.9)	3 (6.1)	0 (0.0)	
The Lawton Instrumental Activities of Daily Living Scale (IADL)				0.058
Completely preserved, *n* (%)	58 (76.3)	35 (71.4)	23 (85.2)	
Moderate disability, *n* (%)	9 (11.8)	5 (10.2)	4 (14.8)	
Strong disability, *n* (%)	9 (11.8)	9 (18.4)	0 (0.0)	
Barthel scale	*n* = 68			0.142
Light, *n* (%)	60 (88.2)	35 (83.3)	25 (96.2)	
Severe, *n* (%)	8 (11.8)	7 (16.7)	1 (3.8)	

**Table 5 ijerph-20-02065-t005:** Number (percentage) of patients in groups differing in age and recommendation for CGA, the result of the independence test (*p*), and the odds ratio and its 95% confidence interval.

Test Results	Recommendation to CGA		*p*	OR [95% CI]
	Yes	No		
Risk of malnutrition or malnutrition	15 (32.6)	3 (11.1)	0.040	3.87 [1.00; 14.9]
Low risk of malnutrition	31 (67.4)	24 (88.9)		1.00 [ref.]
MMSE < 27 points	28 (58.3)	8 (29.6)	0.029	3.32 [1.22; 9.09]
MMSE ≥ 27 points	20 (41.7)	19 (70.4)		1.00 [ref.]
TFI ≥ 4 points	24 (80.0)	5 (26.3)	<0.001	11.2 [2.88; 43.5]
TFI < 4 points	6 (20.0)	14 (73.7)		1.00 [ref.]
ADL < 6 points	16 (32.7)	1 (3.7)	0.004	12.6 [1.57; 101]
ADL ≥ 6 points	33 (67.4)	26 (96.3)		1.00 [ref.]
IADL < 27 points	36 (73.5)	10 (37.0)	0.003	4.71 [1.72; 12.9]
IADL ≥ 27 points	13 (26.5)	17 (63.0)		1.00 [ref.]
Fall risk < 8 points	29 (58.0)	8 (29.6)	<0.001	15.0 [4.67; 48.3]
Fall risk ≥ 8 points	21 (42.0)	19 (70.4)		1.00 [ref.]
Barthel scale < 100 points	29 (69.0)	7 (26.9)	0.031	3.28 [1.21; 8.91]
Barthel scale = 100 points	13 (31.0)	19 (73.1)		1.00 [ref.]
Homocysteine ≥ 10.68 mmol/L	28 (84.8)	15 (62.5)	0.067	3.36 [0.95; 11.8]
Homocysteine < 10.68 mmol/L	5 (15.2)	9 (37.5)		1.00 [ref.]
Sleep duration < 6.6 h a day	12 (48.0)	5 (25.0)	0.203	2.77 [0.77; 9.97]
Sleep duration ≥ 6.6 h a day	13 (52.0)	15 (75.0)		1.00 [ref.]

## Data Availability

Not applicable.

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
