# Peer review of "Identification of Factors Affecting the Increased Percentage of CGA Recommendations among Patients on Geriatric Ward"

_ijerph, 2023, doi:10.3390/ijerph20032065_

Round 1

Reviewer 1 Report (Previous Reviewer 1)

Thank you for answering all my questions.

Author Response

Thank you for providing insightful feedback on ways to strengthen our paper.

Reviewer 2 Report (New Reviewer)

Comprehensive Geriatric Assessment (CGA) is defined as a multimodal and multidisciplinary process which identifies medical, social and functional needs, and the development of an personalized  care plan to older people. CGA is associated with lower mortality and less institutionalization.The goal of a CGA is to identify the geriatric person's needs and is considered the gold standard for managing the frailty.T

The significance of this problem is crucial for both people and globally. Assessment of the overall condition in the elderly can help clinicians identify patients at increased risk of poor outcomes, as well as suggest therapies to maintain and preserve functional independence, reduce disability and improve survival.

The title is informative enough to draw its potential reader's attention. The introduction of this article successfully explains why the understanding the factors increasing the percentage of CGA recommendations among geriatric patients seems to be crucial for health professionals, whose role is to control the aging processes and implement treatment strategies for comorbid diseases, as well as encourage geriatric patients to change their lifestyle and nutrition habits.

In the introduction/background section of this article, successfully explained why current research is important. But the authors could provide some small additional information related to the problem and include additional relevant references.

Here are some minor suggestions that would improve the introduction:

1.               According to the context in the Introduction, the authors can give a scheme/figure of the basic principles/elements or dimensions of the assessment in CGA (physical health, socio-economic status, mental and nutritional status…etc.).

2.               You can give some information about the most commonly used tools in CGA (ARTHEL, IADL. ADLs, MMSE, etc.), the clinical domains, range. If the authors follow the concept of the article, they even can represent these data in table.

3.               According to context, the authors can provide some information worldwide for this problem by giving some other countries as an examples and with relevant references.

In the methodology section, this article is the methods adequately described.

The results clearly presented and illustrated with tables.

The discussion provides specific information on the issue of CGA assessments and shows a comparability of results with other studies in the field. This chapter reveals that these people often have complex, multiple and interdependent problems, which makes them, care more challenging than those with one medical problem. The strength of CGA lies in the fact that it is a multidimensional holistic assessment of adults who takes into consideration health and well-being.

The conclusion section, on the other hand, can be supplemented by an authors’ proposal for the combination of which tools would be most useful for a recommended CGA assessment. Could it be introduced as a mandatory for the elderly over 70 years of age in clinical practice?

Thanks to the authors.

Author Response

Thank you for providing insightful feedback on ways to strengthen our paper. It is with great pleasure that we resubmit our article for further consideration. We have incorporated changes that reflect the detailed suggestions you have graciously provided. We also hope that our edits and the responses we provide below satisfactorily address all the issues and concerns you have noted.

This manuscript is a resubmission of an earlier submission. The following is a list of the peer review reports and author responses from that submission.

Round 1

Reviewer 1 Report

Dear Author

The manuscript is quite interesting but there are some comments to be clarified.

1. How do you calculate the sample size? Did you take all the patients admitted in geriatric ward? If so, what was the duration of study?

2. What is the design of our study?

3. What is the inclusion and exclusion criteria? The methodology should be in detail focusing in all these issues.

4. In table 1, you mentioned that 69 patients are still living? "Still living" indicates till which period? How long you followed the patient's status?

5. In table 4, you considered all studied patients and found out the recommendation to CGA. What is the use in recommending for patients who are not alive? Will that not affect the percentage of recommendation?

6. You mentioned the abbreviation of CGA in the introduction. So avoid elaborating it again in the rest of the manuscript.

7. In table 4, 1st column - risk of malnutrition was categorized into - low risk, high risk and malnutrition. It should be "No malnutrition".

8. In table 5, you mentioned about homocysteine levels. Is it a routine investigation for all the patients? Or was it done for your study? If it was done for your study, mention in detail in the methodology.

Reviewer 2 Report

The authors have put great effort in this study, however there are too many shortcomings and therefore my recommendation is rejection.

The aim of the study is not clearly described: the title and last paragraph of the introduction in which the aim is stated do not match. The participant recruitment lacks many important pieces of information (who were recruited, where when and how), and the measures are inadequately described. References are missing. There is no information on statistical analysis. There is a lot of material in the results-section, however it is very difficult to understand what has been done and why, due to missing information in the methods. It is also impossible to evaluate the results, as there is very little background information on the participants (ses, comorbidities, health status).
